# Kendallknight: An R package for efficient implementation of Kendall's correlation coefficient computation

**Mauricio Vargas Sepulveda** [1,2]*

1 Munk School of Global Affairs and Public Policy, University of Toronto, Toronto, Ontario, Canada,
2 Department of Political Science, University of Toronto, Toronto, Ontario, Canada

* m.sepulveda@mail.utoronto.ca

## Abstract

The kendallknight package introduces an efficient implementation of Kendall's correlation coefficient computation, significantly improving the processing time for large datasets without sacrificing accuracy. The kendallknight package, following Knight (1966) and posterior literature, reduces the time complexity resulting in drastic reductions in computation time, transforming operations that would take minutes or hours into milliseconds or minutes, while maintaining precision and correctly handling edge cases and errors. The package is particularly advantageous in econometric and statistical contexts where rapid and accurate calculation of Kendall's correlation coefficient is desirable. Benchmarks demonstrate substantial performance gains over the Base R implementation, especially for large datasets.

## Introduction

Kendall's correlation coefficient is a non-parametric measure of association between two variables. It is particularly useful when the relationship is monotonic but not necessarily linear, and when data include outliers or ordinal scales.

The implementation in base R has a time complexity of $O(n^2)$, which becomes slow for large datasets [1]. This can introduce bottlenecks when using econometrics or machine learning methods in fields such as genomics or finance, where datasets commonly contain thousands of observations.

Similar to Pearson's correlation, Kendall's implementation in base R uses a multi-threaded implementation and, as the benchmarks reveal, the computational complexity still constitutes a bottleneck even with top-of-the-line hardware. Alternative implementations, such as Python's SciPy have a computational complexity of $O(n \ln(n))$ that scale well with non-trivial missing data patterns, ties, or corner cases.

To address this, we implemented a high-performance version of Kendall's $\tau$ in the `kendallknight` R package using C++, building on the algorithm introduced by Knight [2], refined in subsequent work [3,4], and following programming principles from [5]. Our approach achieves $O(n \ln(n))$ time complexity, which represents a substantial reduction in computational cost. For example, with $n = 20,000$ observations, an $O(n^2)$ method

**Data availability statement:** The data can be found in https://github.com/pachadotdev/spuriouscorrelations and https://github.com/pachadotdev/tradepolicy.

**Funding:** The author(s) received no specific funding for this work.

**Competing interests:** The authors have declared that no competing interests exist.

requires roughly 400 million pairwise comparisons, while our implementation completes in under 200,000 operations.

This efficiency gain translates into practical improvements: in benchmark tests on real-world datasets, we observe reductions in execution time of several minutes of over $10,000\%$, without loss of precision or robustness. We also include comprehensive unit tests to validate correctness across edge cases, including tied ranks and degenerate inputs.

In summary, this package provides a fast, reliable, and scalable alternative for computing Kendall's correlation, with applications across fields where large-scale non-parametric correlation analysis is needed.

## Definitions

Kendall's correlation coefficient is a pairwise measure of association. It does not require assumptions about the distribution of the data (e.g., normality), and is especially appropriate for ordinal data or data with outliers, where linear correlation measures like Pearson's may be misleading. For two vectors $x$ and $y$ of length $n$, it is defined as [2]:

$$r(x, y) = \frac{c - d}{\sqrt{(c + d + e)(c + d + f)}},$$

where $c$ is the number of concordant pairs, $d$ is the number of discordant pairs, $e$ is the number of ties in $x$, and $f$ is the number of ties in $y$.

The corresponding definitions for $c$, $d$, $e$ and $f$ are:

$$c = \sum_{i=1}^{n} \sum_{j \neq i}^{n} g_1(x_i, x_j, y_i, y_j),$$

$$d = \sum_{i=1}^{n} \sum_{j \neq i}^{n} g_2(x_i, x_j, y_i, y_j),$$

$$e = \sum_{i=1}^{n} \sum_{j \neq i}^{n} g_3(x_i, x_j) g_4(y_i, y_j),$$

$$f = \sum_{i=1}^{n} \sum_{j \neq i}^{n} g_4(x_i, x_j) g_3(y_j, y_i).$$

The functions $g_1$, $g_2$, $g_3$ and $g_4$ are indicators defined as:

$$g_1(x_i, x_j, y_i, y_j) = \begin{cases} 1 & \text{if } (x_i - x_j)(y_i - y_j) > 0, \\ 0 & \text{otherwise,} \end{cases}$$

$$g_2(x_i, x_j, y_i, y_j) = \begin{cases} 1 & \text{if } (x_i - x_j)(y_i - y_j) < 0, \\ 0 & \text{otherwise,} \end{cases}$$

$$g_3(x_i, x_j) = \begin{cases} 1 & \text{if } x_i = x_j \text{ and } y_i \neq y_j, \\ 0 & \text{otherwise,} \end{cases}$$

$$g_4(y_i, y_j) = \begin{cases} 1 & \text{if } x_i \neq x_j \text{ and } y_i = y_j, \\ 0 & \text{otherwise.} \end{cases}$$

Kendall's coefficient reflects the difference between the number of concordant and discordant pairs, normalized by a correction factor to account for ties. The total number of comparisons is $m = n(n-1)/2$, so a naive implementation that checks all pairs has a time complexity of $O(n^2)$.

When there are no ties in the data, the coefficient simplifies to:

$$r(x,y) = \frac{c-d}{c+d} = \frac{c-d}{m} = \frac{4c}{n(n-1)} - 1$$

Although this formula is straightforward, computing it directly is inefficient for large datasets. Instead, efficient implementations use sorting and inversion algorithms, a method borrowed from merge sort algorithms in binary trees, to compute $c$ and $d$ with time complexity $O(n\ln(n))$ [2].

Unlike the Pearson's correlation coefficient, suitable for continuous variables and defined as

$$r(x,y) = \frac{n\sum_{i=1}^{n} x_i y_i - \sum_{i=1}^{n} x_i \sum_{i=1}^{n} y_i}{\sqrt{(n\sum_{i=1}^{n} x_i^2 - (\sum_{i=1}^{n} x_i)^2)(n\sum_{i=1}^{n} y_i^2 - (\sum_{i=1}^{n} y_i)^2)}},$$

the Kendall's correlation coefficient is suitable for ordinal variables.

While Pearson's correlation coefficient measures linear relationships and is sensitive to outliers, non-parametric alternatives like Kendall's $\tau$ assess monotonic relationships and are more robust.

Because of these properties, Kendall's correlation is often preferred in social sciences, bioinformatics, and ordinal regression diagnostics, where the goal is to detect reliable monotonic associations without assuming a functional form.

## Implementation

Using a merge sort with a binary tree with depth $1 + \log_2(n)$ results in a search and insert operation with a time complexity of $O(\log(n))$, resulting in a time complexity of $O(n\log(n))$ for the Kendall's correlation coefficient [2,5].

To address this, `kendallknight` implements an algorithm that reduces the computational complexity to $O(n\ln(n))$ by leveraging merge sort and a binary indexed tree (Fenwick tree). As originally proposed by [2], this approach counts the number of inversions in the rank-transformed vector $y$ after sorting $x$.

The `kendallknight` algorithm consists of the following high-level steps:

1. Sort the vector $x$ and keep track of the original indices in a permutation vector.
2. Rearrange the vector $y$ according to $x$.
3. Compute the total pairs $m$.
4. Compute the pairs of ties in $x$ as $m_x = t_x(t_x + 1)/2$.
5. Compute the pairs of ties in $y$ as $m_y = t_y(t_y + 1)/2$.
6. Compute the concordant pairs adjusted by the number of swaps in $y$ by using a merge sort as $t = m - t_x - t_y + 2t_p$.
7. Compute the Kendall's correlation coefficient as $r(x,y) = t/(\sqrt{m - m_x}\sqrt{m - m_y})$.

The `kendallknight` package implements these steps in C++ and exports the Kendall's correlation coefficient as a function that can be used in R by using the `cpp11` headers [6]. Unlike existing implementations with $O(n\log(n))$ complexity, this implementation also

provides dedicated functions to test the statistical significance of the computed correlation, and for which it uses a C++ port of the Gamma function that R already implemented in C [1,7].

Below is pseudocode summarizing the core logic implemented in C++:

```
function kendall_tau(x, y):
    n ← length(x)
    x_ranked ← rank(x)
    y_ranked ← rank(y)
    pairs ← sort (x_ranked[i], y_ranked[i])
        for i in 1..n by x_ranked
    BIT ← empty binary indexed tree
    discordant ← 0

    for i from 1 to n:
        y_val ← pairs[i].y
        discordant += BIT.query_range(y_val + 1, n)
        BIT.update(y_val)

    m ← n(n - 1) / 2
    concordant ← m - discordant
    tie_x ← count_tied_pairs(x_ranked)
    tie_y ← count_tied_pairs(y_ranked)

    denominator ← sqrt((m - tie_x) * (m - tie_y))
    tau ← (concordant - discordant) / denominator

    return tau
```

While the package provides two user-visible functions detailed in the next section, the `kendall_cor()` and `kendall_cor_test()` functions, which depend on internal functions to compute the Kendall's correlation coefficient and the *p*-value of the test efficiently, which required to port some R methods implemented in C to C++ to avoid the overhead of copying data between the two languages multiple times. Fig 1 shows the data flow in the package:

The `check_data()` function ensures that the input vectors (or matrices) *x* and *y* are suitable for analysis by performing several checks and preparations. It checks that the inputs have the same dimensions, removes missing values, ensures there are at least two non-null observations, and checks for zero variance. If all checks pass, it assigns the cleaned and ranked data to the parent environment and returns `TRUE`. Otherwise, it returns `FALSE` or stops with an error message.

The `insertion_sort()` function performs an insertion sort on an array of doubles (average complexity $O(n^2)$). It iterates through the array, comparing each element with the next one and shifting elements to the right until the correct position for the current element is found. The function keeps track of the number of swaps made during the sorting process and returns this count. This sorting algorithm is efficient for small arrays and is used as a helper function in larger sorting operations.

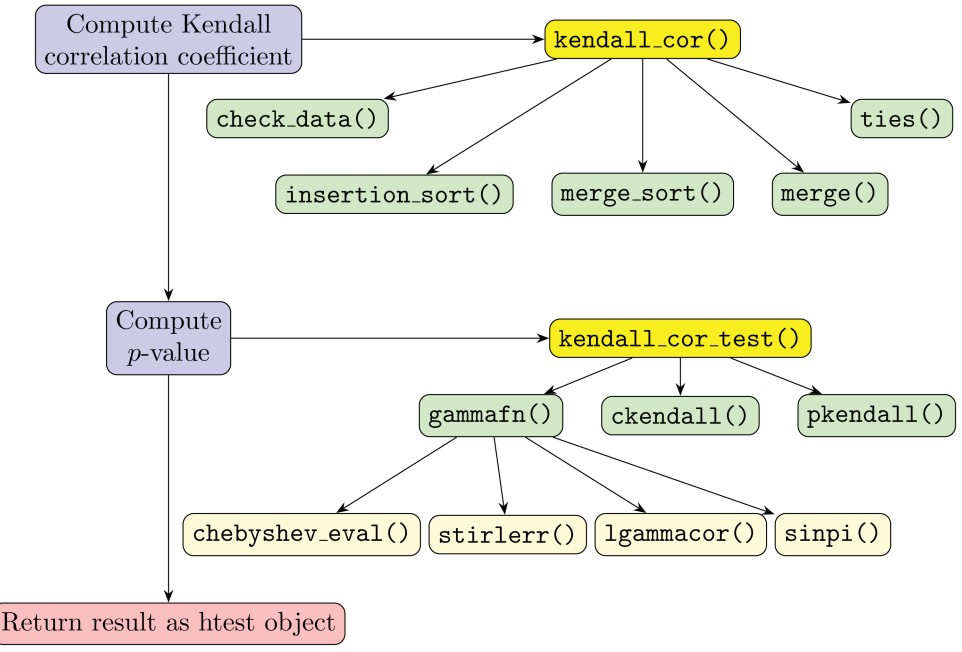

**Fig 1. Workflow diagram showing the data flow in the kendallknight package.** Source: own creation.

The `merge_sort()` function performs a merge sort on an array of doubles (complexity $O(n \log(n))$). If the length of the array is less than 10, it uses the `insertion_sort()` function for sorting, as insertion sort is more efficient for small arrays. Otherwise, it recursively divides the array into two halves, sorts each half using merge_sort_, and then merges the sorted halves using the `merge()` function. The function keeps track of the number of swaps made during the sorting process and returns this count. After merging, it copies the sorted elements back into the original array.

The `merge()` function merges two sorted subarrays into a single sorted array (complexity $O(n)$). It takes two pointers, left and right, pointing to the beginning of each subarray, and iteratively compares the elements at these pointers. The smaller element is copied to the to array, and the corresponding pointer is advanced. This process continues until all elements from one subarray are copied. Any remaining elements from the other subarray are then copied to the to array. The function also keeps track of the number of swaps made during the merge process and returns this count.

The `ties()` function calculates the number of tied pairs in a sorted array of doubles (complexity $O(n)$). It iterates through the array, counting ties or consecutive equal elements. When a tie sequence ends, it calculates the number of tied pairs as $t_z(t_z + 1)/2$. This process continues until the end of the array, ensuring all ties are accounted for.

The `gammafn()` function computes the gamma function value for a given input $x$. It handles special cases (e.g., null, zero, and negative values). For inputs less than or equal to ten, it uses a Chebyshev series evaluation to compute the gamma function. For larger inputs, it uses an approximation involving logarithms and exponential functions. The function also includes corrections for precision and range errors, and handles special cases for very large (or small) inputs. If $x$ is positive, it returns the computed gamma value. If $x$ is negative, it uses the reflection formula involving the sine function to compute the gamma value.

The `ckendall()` function computes the Cumulative Distribution Function (CDF) for Kendall's correlation (or tau statistic) using recursion. It takes three parameters: (1) *k*, the value of the statistic; (2) *n*, the number of observations; and (3) *w*, a memoization table to store intermediate results. The function first calculates the maximum possible value of the statistic as *u*. If *k* is outside the valid range, it returns zero. If the value for *w*(*n*,*k*) has not been computed yet (indicated by a negative value), it recursively computes the CDF by summing the results of smaller subproblems. The results are stored in the memoization table to avoid redundant calculations. The function uses OpenMP (when available) for parallelization to speed up the computation. Finally, it returns the computed CDF value for the given *k* and *n*.

The `pkendall()` function computes the *p*-values for Kendall's correlation for a given vector of test statistics *Q* and the number of observations *n*. It initializes a memoization table *w* to store intermediate results for the CDF calculations. For each element in *Q*, it checks if the value is outside the valid range and assigns the corresponding *p*-value as zero or one. For valid values, it computes the CDF by summing the results of the `ckendall()` function for all values up to the given statistic, normalizing the result by dividing by the gamma function of *n*+1. The function uses OpenMP (if available) for parallelization to speed up the computation. Finally, it returns a vector of *p*-values corresponding to the input test statistics.

The package uses `testthat` for testing [8]. The included tests are exhaustive and covered the complete code to check for correctness comparing with the Base R implementation, and also checking corner cases and forcing errors by passing unusable input data to the user-visible functions. The current tests cover 100% of the code.

## Usage

The `kendallknight` package is exclusively focused on the Kendall's correlation coefficient and provides additional functions to test the statistical significance of the computed correlation not available in other packages, which is particularly useful in econometric and statistical contexts.

The `kendallknight` package is available on CRAN and can be installed using the following command:

```
# CRAN
install.packages("kendallknight")

# GitHub
remotes::install_github("pachadotdev/kendallknight")
```

As an illustrative exercise we can explore the question 'is there a relationship between the number of computer science doctorates awarded in the United States and the total revenue generated by arcades?' Certainly, this question is about a numerical exercise and not about causal mechanisms.

Table 1 can be used to illustrate the usage of the `kendallknight` package:

The `kendall_cor()` function can be used to compute the Kendall's correlation coefficient:

```
library(kendallknight)

kendall_cor(arcade$doctorates, arcade$revenue)
```

**Table 1. Computer science and arcade revenue dataset. Source: [9].**

|      | Computer science doctorates awarded in the US | Total revenue generated by arcades |
|------|-----------------------------------------------|-------------------------------------|
| 2000 | 861  | 1.196 |
| 2001 | 830  | 1.176 |
| 2002 | 809  | 1.269 |
| 2003 | 867  | 1.240 |
| 2004 | 948  | 1.307 |
| 2005 | 1129 | 1.435 |
| 2006 | 1453 | 1.601 |
| 2007 | 1656 | 1.654 |
| 2008 | 1787 | 1.803 |
| 2009 | 1611 | 1.734 |

```
[1] 0.8222222
```

The `kendall_cor_test()` function can be used to test the null hypothesis that the Kendall's correlation coefficient is zero:

```
kendall_cor_test(
  arcade$doctorates,
  arcade$revenue,
  conf.level = 0.8,
  alternative = "greater"
)
```

```
Kendall's rank correlation tau

data:  arcade$doctorates and arcade$revenue
tau = 0.82222, p-value = 0.0001788
alternative hypothesis: true tau is greater than 0
80 percent confidence interval:
 0.5038182 1.0000000
```

One important difference with base R implementation is that this implementation allows to obtain confidence intervals for different confidence levels (e.g., 95%, 90%, etc).

With the obtained $p$-value and a significance level of 80% (the default is 95%), the null hypothesis is rejected for the two-tailed test ($H_0 : \tau = 0$ versus $H_1 : \tau \neq 0$, the default option) and the greater than one-tailed test ($H_0 : \tau = 0$ versus $H_1 : \tau > 0$) but not for the lower than one-tailed test ($H_0 : \tau = 0$ versus $H_1 : \tau < 0$). This suggests the correlation is positive (e.g., more doctorates are associated with more revenue generated by arcades). In other words, these three tests tell us that the empirical evidence from this dataset provides three answers to the research questions:

1. Is there any relationship? Yes, more doctorates are associated with more revenue generated by arcades.
2. Is there a positive relationship? Yes, more doctorates are associated with more revenue generated by arcades.

3. Is there a negative relationship? No, more doctorates are not associated with less revenue generated by arcades.

With base R or `Kendall`, an equivalent result can be obtained with the following code:

```
cor.test(arcade$doctorates, arcade$revenue, method = "kendall")
```

```
Kendall's rank correlation tau

data: arcade\textit{doctorates}\textit{and}\textit{arcade}revenue
T = 41, p-value = 0.0003577
alternative hypothesis: true tau is not equal to 0
sample estimates:
tau
```

```
Kendall::Kendall(arcade$doctorates, arcade$revenue)
```

```
 tau = 0.822, 2-sided pvalue =0.0012822
```

In an Econometric context, the current implementation is particularly useful to compute the pseudo-$R^2$ statistic defined as the squared Kendall correlation in the context of (Quasi) Poisson regression with fixed effects [10,11]. A local test reveals how the pseudo-$R^2$ computation time drops from fifty to one percent of the time required to compute the model coefficients by using the `fepois()` function from the `lfe` package [12] and a dataset containing fifteen thousand rows [13]:

```
library(tradepolicy)
library(lfe)

data8694 <- subset(agtpa_applications, year %in% seq(1986, 1994, 4))

fit <- fepois(
  trade ~ dist + cntg + lang + clny + rta |
    as.factor(paste0(exporter, year)) +
    as.factor(paste0(importer, year)),
  data = data8694
)

psr <- (cor(data8694$trade, fit$fitted.values, method = "kendall"))^2

psr2 <- (kendall_cor(data8694$trade, fit$fitted.values))^2
```

The percentages in Table 2 reveal that base R implementation takes around 50% of the time required to fit the model to compute the pseudo-$R^2$ statistic, while the `kendallknight` implementation takes only 1% of the model time.

## Benchmarks

We tested the `kendallknight` package against the base R implementation of the Kendall correlation using the `cor` function with `method = "kendall"` for randomly generated

**Table 2. Computation time for the model coefficients and the pseudo-$R^2$ statistic. Source: own creation.**

| Operation | Time | Pseudo $R^2$/Model fitting |
|---|---|---|
| Model fitting | 3.75 s | |
| Pseudo-$R^2$ (base R) | 1.78 s | 47.58% |
| Pseudo-$R^2$ (kendallknight) | 0.02 s | 0.51% |

vectors of different lengths, and against the `Kendall` package [14]. The data used for the benchmark is the trade panel available in [13].

We used the `bench` package to run the benchmarking tests in a clean R session in the Niagara supercomputer cluster that, unlike personal computers, will not distort the test results due to other processes running in the background (e.g., such as automatic updates).

This cluster has the following specifications:

- Nodes: 2,024 compute nodes
- Processors: Each node equipped with dual Intel Xeon Skylake (2.4 GHz) or Cascade Lake (2.5 GHz) CPUs, totaling 40 cores per node
- Memory: 202 GB RAM per node
- Storage: 12.5 PB scratch space, 3.5 PB project space, and a 256 TB burst buffer
- Operating System: CentOS 7

Due to the nature of this benchmark, we used one node (40 cores).

The values on Table 3 reveal that the `kendallknight` package is orders of magnitude faster than `Kendall` and the base R implementation for large datasets.

The values on Table 4 show that `kendallknight` has a marginal memory allocation overhead compared to the base R implementation. The same applies to the `Kendall` package.

## Conclusion

The `kendallknight` package provides a fast and memory-efficient implementation of the Kendall's correlation coefficient with a time complexity of $O(n \log(n))$, which is orders of magnitude faster than the base R implementation without sacrificing precision or correct handling of corner cases. Pearson's and Spearman's correlation coefficients were not considered as base R already provides efficient implementations of these methods.

**Table 3. Computation time by number of observations. Source: own creation.**

| No. of observations | kendallknight median time (s) | Kendall median time (s) | Base R median time (s) |
|---|---|---|---|
| 10,000 | 0.013 | 1.0 | 4 |
| 20,000 | 0.026 | 3.9 | 16 |
| 30,000 | 0.040 | 8.7 | 36 |
| 40,000 | 0.056 | 15.6 | 64 |
| 50,000 | 0.071 | 24.2 | 100 |
| 60,000 | 0.088 | 34.8 | 144 |
| 70,000 | 0.104 | 47.5 | 196 |
| 80,000 | 0.123 | 61.9 | 256 |
| 90,000 | 0.137 | 78.2 | 324 |
| 100,000 | 0.153 | 96.4 | 399 |

**Table 4. Memory allocation by number of observations. Source: own creation.**

| No. of observations | kendallknight memory allocation (MB) | Kendall memory allocation (MB) | Base R memory allocation (MB) |
|---|---|---|---|
| 10,000 | 1.2 | 1.1 | 0.89 |
| 20,000 | 2.3 | 2.1 | 1.60 |
| 30,000 | 3.5 | 3.1 | 2.40 |
| 40,000 | 4.6 | 4.2 | 3.20 |
| 50,000 | 5.8 | 5.2 | 4.00 |
| 60,000 | 7.0 | 6.2 | 4.80 |
| 70,000 | 8.1 | 7.3 | 5.60 |
| 80,000 | 9.3 | 8.3 | 6.40 |
| 90,000 | 10.4 | 9.4 | 7.20 |
| 100,000 | 11.6 | 10.4 | 8.00 |

The current implementation does not leverage multi-threading or parallel computing for all operations, which could further enhance performance on multi-core systems. This is an area for future development, as the current implementation is already significantly faster than the base R implementation and the `Kendall` package.

For small vectors (e.g., less than 100 observations), the time difference is negligible. However, for larger vectors, the difference can be substantial. This package is particularly useful to solve bottlenecks in the context of econometrics and international trade, but it can also be used in other fields where the Kendall's correlation coefficient is required.

The software, documentation, and replication code are available on GitHub.

## Author contributions

**Conceptualization:** Mauricio Vargas Sepulveda.

**Data curation:** Mauricio Vargas Sepulveda.

**Formal analysis:** Mauricio Vargas Sepulveda.

**Software:** Mauricio Vargas Sepulveda.

**Validation:** Mauricio Vargas Sepulveda.

**Writing – original draft:** Mauricio Vargas Sepulveda.

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
