## [Decision Letter · Decision Letter 0]

27 Apr 2025

PONE-D-25-15032Kendallknight: An R Package for Efficient Implementation of Kendall’s Correlation Coefficient ComputationPLOS ONE

Dear Dr. Vargas,

Thank you for submitting your manuscript to PLOS ONE. After careful consideration, we feel that it has merit but does not fully meet PLOS ONE’s publication criteria as it currently stands. Therefore, we invite you to submit a revised version of the manuscript that addresses the points raised during the review process.

 All the reviewers have found that the manuscript has relevant contribution to the topic of computational statistics and data analysis. However, two reviewers have pointed out some important points that would require revision by the author. One of these reviewers has attached a document (in Word format) with all the details of these points raised that require careful attention by the author in a revised response/submission. ==============================

We look forward to receiving your revised manuscript.

Kind regards,

Carlos Eduardo Thomaz, Ph.D.

Academic Editor

PLOS ONE

Journal Requirements:

kendallknight - https://pacha.dev/kendallknight/

In your revision ensure you cite all your sources (including your own works), and quote or rephrase any duplicated text outside the methods section. Further consideration is dependent on these concerns being addressed.

Additional Editor Comments:

All the reviewers have found that the manuscript has relevant contribution to the topic of computational statistics and data analysis. However, two reviewers have pointed out some important points that would require revision by the author.

Reviewers' comments:

Reviewer's Responses to Questions

**Comments to the Author**

1. Is the manuscript technically sound, and do the data support the conclusions?

Reviewer #1: Yes

Reviewer #2: Partly

Reviewer #3: Yes

2. Has the statistical analysis been performed appropriately and rigorously? 

Reviewer #1: Yes

Reviewer #2: Yes

Reviewer #3: Yes

3. Have the authors made all data underlying the findings in their manuscript fully available?

Reviewer #1: Yes

Reviewer #2: No

Reviewer #3: Yes

4. Is the manuscript presented in an intelligible fashion and written in standard English?

Reviewer #1: Yes

Reviewer #2: Yes

Reviewer #3: Yes

5. Review Comments to the Author

Reviewer #1: The reviewer endorsese the publication of the present manuscript. In this manuscript author/s explains the so-called The kendallknight package that introduces an efficient implementation of Kendall’s correlation coefficient computation, significantly improving the processing time for large datasets without sacrificing accuracy. Furthermore, following Knight (1966) and posterior literature, author/s affirms that this new type of R package is able to reduce the computational complexity deriving from the drastic reductions in computation time.- In this manner it will transform operations that would take minutes or hours into milliseconds or minutes, while maintaining precision and correctly

handling edge cases and errors.

Reviewer #2: Dear Authors

It was a pleasure reading your work. It is an interesting study, but requires minor improvements before it can be accepted for publication to maximize the article's impact.

Please find them in the attached file.

All the best.

Reviewer #3: The manuscript "Kendallknight: An R Package for Efficient Implementation of Kendall's Correlation Coefficient Computation" introduces an R package that computes Kendall’s correlation coefficient with a significant improvement over base R’s implementation. It demonstrates robust performance through benchmarks.I recommend acceptance with minor modifications:

1. Clarify the claim about base R’s (O(n^2))) complexity with reference that could strengthen this.

2. Briefly explain the Kendall correlation output -0.4925183 for non-expert readers.

6. PLOS authors have the option to publish the peer review history of their article (what does this mean?). If published, this will include your full peer review and any attached files.

Reviewer #1: No

Reviewer #2: No

Reviewer #3: No

---

## [Author Response · Author response to Decision Letter 1]

16 May 2025

Dear Reviewers,

Thanks for the comments. I consider each of them as I considered these improve the draft.

I have re-written the article based on the received feedback.

However, I do not have a track changes feature in Quarto/Latex. I added a new rendered PDF.

---

## [Decision Letter · Decision Letter 1]

26 May 2025

Kendallknight: An R Package for Efficient Implementation of Kendall’s Correlation Coefficient Computation

PONE-D-25-15032R1

Dear Dr. Vargas,

We’re pleased to inform you that your manuscript has been judged scientifically suitable for publication and will be formally accepted for publication once it meets all outstanding technical requirements.

Kind regards,

Carlos Eduardo Thomaz, Ph.D.

Academic Editor

PLOS ONE

Additional Editor Comments (optional):

All the main points raised by the reviewers have been properly addressed in this new version of the manuscript. Therefore, no further revision is needed and the new version of the manuscript can be accepted as it is.

Reviewers' comments:

Reviewer's Responses to Questions

**Comments to the Author**

1. If the authors have adequately addressed your comments raised in a previous round of review and you feel that this manuscript is now acceptable for publication, you may indicate that here to bypass the “Comments to the Author” section, enter your conflict of interest statement in the “Confidential to Editor” section, and submit your "Accept" recommendation.

Reviewer #1: All comments have been addressed

Reviewer #2: All comments have been addressed

2. Is the manuscript technically sound, and do the data support the conclusions?

Reviewer #1: Yes

Reviewer #2: Yes

3. Has the statistical analysis been performed appropriately and rigorously? 

Reviewer #1: Yes

Reviewer #2: Yes

4. Have the authors made all data underlying the findings in their manuscript fully available?

Reviewer #1: Yes

Reviewer #2: Yes

5. Is the manuscript presented in an intelligible fashion and written in standard English?

Reviewer #1: Yes

Reviewer #2: Yes

6. Review Comments to the Author

Reviewer #1: The reviewer endorses the publication of the present manuscript because the athour/s have completely and adequately addressed the reviewer comments raised in a previous round of evaluation.

Reviewer #2: Dear Authors,

Thank you for your thoughtful and comprehensive revision. You have addressed the reviewer’s suggestions.

Congratulations on a strong and well-executed revision.

All the best!

7. PLOS authors have the option to publish the peer review history of their article (what does this mean?). If published, this will include your full peer review and any attached files.

Reviewer #1: **Yes: **Daniela Cialfi

Reviewer #2: No

---

## [Editor Report · Acceptance letter]

PONE-D-25-15032R1

PLOS ONE

Dear Dr. Sepulveda,

I'm pleased to inform you that your manuscript has been deemed suitable for publication in PLOS ONE. Congratulations! Your manuscript is now being handed over to our production team.

Kind regards,

on behalf of

Prof. Carlos Eduardo Thomaz

Academic Editor

PLOS ONE